# OpenReview forum: "Synthetic Data (Almost) from Scratch: Generalized Instruction Tuning for Language Models"
_NeurIPS.cc/2024/Conference — Submitted to NeurIPS 2024_

### Official Review · Reviewer_oEvn · 2024-07-10

**Soundness:** 3
**Presentation:** 3
**Contribution:** 3
**Rating:** 7
**Confidence:** 4

**Summary:**

The paper introduces generalized isntruction tuning (GLAN), an approach for synthesizing instruction tuning data using a taxonomy-based approach. GLAN generates synthetic instruction data from pre-curated taxonomy of human knowledge and capabilities and aims to create diverse and broad-ranging instruction dataset.

**Strengths:**

1. Comprehensive Coverage of Evaluation: The paper presents extensive experiments demonstrating that GLAN outperforms various popular instruction-tuned LLMs across multiple dimensions, including mathematical reasoning, coding, logical reasoning, and general instruction following.
2. Minimization of Human Involvement: The generation process significantly reduces human involvement, requiring human verification only at the taxonomy construction stage. This makes the approach scalable and less labor-intensive.
3. Customizability and Extensibility: The taxonomy-based approach allows for easy customization and extension. New fields or skills can be incorporated by simply adding new nodes to the taxonomy.

**Weaknesses:**

1. While the paper addresses generalization, there is a risk that the generated synthetic data might overfit to the taxonomy's structure, potentially missing out on more nuanced, real-world instructions.

**Questions:**

1. Are there any measures in place to ensure the generated synthetic data's diversity and prevent redundancy?
2. what is the whole taxonomy of the human knowledge and capabilities? And can each task (e.g., gsm8k, arc) be categorised into any sub category?
3. Are the effects of each category orthogonal to each other? i.e., ablating data from a child category does not effect tasks in another child category. It would be beneficial if the authors could provide some preliminary results.

**Limitations:**

The authors have adequately addressed the limitations

---

> ### Author Rebuttal · Authors · 2024-08-07
>
> Thanks for your thorough evaluation and insightful feedback on our submission.
>
> ---
> > While the paper addresses generalization, there is a risk that the generated synthetic data might overfit to the taxonomy's structure, potentially missing out on more nuanced, real-world instructions
>
> GLAN is customizable. One can always introduce rare fields/disciplines to the taxonomy to capture nuanced, real-world instruction data as needed. Sometimes, the real-world instructions might be too difficult to “summarize” (into a certain discipline/field) manually. One workaround is to prompt GPT-4 with some of your instructions to generate the field or discipline names. The most frequently occurring field/discipline names can then be incorporated into the taxonomy.
>
> ---
>
> > Are there any measures in place to ensure the generated synthetic data's diversity and prevent redundancy?
>
> Yes. As described in line 161 to 172 (Section 2.4), we incorporate randomly sampled class sessions and key concepts to our instruction generation prompt to ensure the diversity of our generated instructions (also see the full prompt in Table 6 of Appendix A.4). For each discipline, we have approximately 4 million unique combinations (of randomly sampled class sessions and key concepts) and in total (126 disciplines) we have over 500 million such unique combinations, which guarantees that the 10 million instructions we generate exhibit significant diversity.
>
> ---
>
> > what is the whole taxonomy of the human knowledge and capabilities?
>
> As described in line 186 (section 3.1), we uploaded the taxonomy of human knowledge and capabilities used in this work as the supplementary material. Below is an example of the leaf node (i.e., the “History” discipline), where the parent node is the “Humanities” field and the grandparent node is the “academic” category.
> ```
> {"topic": "History", "meta_topic": "academic, Humanities"}
> ```
>
> ---
>
> > And can each task (e.g., gsm8k, arc) be categorised into any sub category?
>
> It depends. Specifically, gsm8k aligns well with the mathematics discipline in our taxonomy. In contrast, tasks like ARC and MMLU span multiple disciplines, corresponding to various nodes within our taxonomy.  It is challenging to determine which specific discipline would most benefit the reasoning-related BBH task. However, after training on the 10 million generated instruction data, we observed a notable improvement in the reasoning capabilities of LLMs.
>
> ---
>
> > Are the effects of each category orthogonal to each other? i.e., ablating data from a child category does not affect tasks in another child category. It would be beneficial if the authors could provide some preliminary results.
>
> According to our ablation experiments on the Mathematics discipline, the answer is yes. We use 15K data generated from the Mathematics discipline (from our taxonomy) and 60K data generated from all the other disciplines (also from our taxonomy). We run experiments in three settings (i.e., only with the 15K math data, only with the 60K data in other disciplines and combining the 15K and the 60K data) and results are as follows.
> |                        | GSM8K | HumanEval | BBH  |
> |------------------------|-------|-----------|------|
> | math 15k               | 60.7 | 35.9      | 57.7 |
> | others 60k             | 31.3  | 41.5      | 58.8 |
> | math 15k + others 60k  | 61.5 | 40.2      | 58.7 |
> |
>
> The results from the GSM8K benchmark indicate that the mathematical capabilities are predominantly derived from the data generated within the Mathematics discipline. When combined with the 60K data from other disciplines, the mathematical performance remains mostly unchanged. Similarly, the coding and reasoning capabilities appear to be derived from the data of other disciplines (see results on HumanEval and BBH), and the same trends are observed when this data is combined with the math data.
> Thank you for bringing this interesting question to us. We will include the above explanations and results into our revised manuscript.
>
> ---
>
> All the results and discussions mentioned above will be included in our updated manuscript.
>
> Thank you again for your constructive reviews.

---

> > ### Comment · Reviewer_oEvn · 2024-08-09
> >
> > Thanks for the authors responses and most of my concerns are properly addressed. Thus I would like to remain the score as 7 (Accept).

---

> > > ### Author Response · Authors · 2024-08-09
> > >
> > > Thank you again for your thorough review and for taking the time to carefully read our responses. We are glad that most of your concerns have been addressed, and we appreciate your continued support for our work.

---

### Official Review · Reviewer_koAL · 2024-07-12

**Soundness:** 3
**Presentation:** 3
**Contribution:** 2
**Rating:** 5
**Confidence:** 3

**Summary:**

This paper introduces GLAN, a general and scalable method for instruction tuning of Large Language Models (LLMs). GLAN employs a top-down approach to generate high-quality instruction tuning datasets. Experiments across various benchmarks demonstrate that GLAN performs comparably to other existing methods.

**Strengths:**

1. This paper focuses on the alignment of Large Language Models, which is a trendy and important topic. If the dataset is released, it will be beneficial for the community.
2. This method is easy to follow. The process is highly scalable, leveraging LLMs like GPT-4 for generating instructions on a massive scale.
GLAN allows for easy customization. New fields can be added by incorporating new nodes into the taxonomy.

**Weaknesses:**

The novelty is limited as similar top-down designs have been utilized in many previous works. Besides, the main experimental results in Table 1 appear mediocre compared to other methods.

**Questions:**

Refer to the weakness.

**Limitations:**

Refer to the weakness.

---

> ### Author Rebuttal · Authors · 2024-08-07
>
> Thank you for taking the time to review our submission and providing constructive feedback.
>
> ---
> > The novelty is limited as similar top-down designs have been utilized in many previous works.
>
> Regarding "similar top-down designs in previous works", could you please share specific references or examples? This would help us further clarify the distinctions between GLAN and those approaches.
>
> Besides, the key contributions of GLAN include:
>
> - **Scalability and Independence from Pre-Existing Data**: As described in Section 2 (see lines 82-95), GLAN enables LLMs such as GPT-4/GPT-4o and GPT-3.5/GPT-4o-mini to generate a vast amount of instruction-response data from scratch, exceeding 500 million data points (i.e., 500 million unique combinations of knowledge points), without relying on pre-existing data samples. This approach contrasts with previous methods like Evolve-Instruct and Self-Instruct, which depend on pre-existing data samples (see lines 29-37). For instance, WizardCoder [1] utilizes 20K provided samples, resulting in a final data size of 80K after applying the Evolve-Instruct method.
>
> - **Broad Coverage Across Disciplines**: GLAN spans a wide range of domains, currently covering 126 disciplines. This extensive coverage ensures that the generated data is diverse and comprehensive. For a detailed overview of the 126 disciplines, please refer to Table 12 in the appendix or the supplementary materials.
>
> - **Easy Customizability**: As detailed in lines 74-77, GLAN offers easy customization options, allowing users to add new fields, disciplines, or subjects into the taxonomy without disrupting previously generated content. This flexibility ensures that GLAN can be adapted to various needs and updated as new knowledge areas emerge.
>
> ---
>
> > Besides, the main experimental results in Table 1 appear mediocre compared to other methods.
>
> Without utilizing task-specific training data of these tasks, we achieved the highest or second-highest results as shown in Table 1 (see the “GLAN” row). Specifically, our approach yielded the best results for MBPP, BBH, ARC-E, ARC-C, and MMLU, and secured the second-best results for HumanEval, GSM8K, and MATH (also see line 249 to 258).  These results demonstrate that after instruction tuning, GLAN excels on multiple dimensions from mathematical reasoning, coding, reasoning, and academic exams with a systematic data generation approach.
>
> Besides these general capabilities as demonstrated on general benchmarks, GLAN also excels on instruction following as shown in Table 4 and Table 5.
>
> ---
> Thanks again for your review!
>
> [1]. Z. Luo et al. “WizardCoder: Empowering Code Large Language Models with Evol-Instruct”. 2023

---

> > ### Author Response · Authors · 2024-08-12
> >
> > Dear Reviewer koAL,
> >
> > Since we are approaching the deadline of the discussion period, we were wondering if you have had the chance to review our response to your comments. We would like to kindly inquire about the extent to which we have successfully addressed the concerns outlined in your review. We greatly value your feedback and would appreciate any further questions or comments you might have.
> >
> > Thank you for your time and consideration.
> >
> >
> > Sincerely,
> >
> > All Authors

---

> > > ### Comment · Reviewer_koAL · 2024-08-12
> > >
> > > Hi,
> > > Thanks for your reply! Regarding the "similar top-down designs in previous works," I was referring to WizardLM and Ultrachat. However, I noticed that your approach distinguishes itself by incorporating a pre-curated taxonomy, which frees it from the constraints of existing data. In light of this, I will be adjusting my score to 5.

---

> > > > ### Author Response · Authors · 2024-08-13
> > > >
> > > > Thanks for taking time to reassess our work and for your updated score!

---

### Official Review · Reviewer_7Qp1 · 2024-07-15

**Soundness:** 3
**Presentation:** 3
**Contribution:** 2
**Rating:** 4
**Confidence:** 4

**Summary:**

This paper proposes a generalized way of creating instruction data. The high-level motivation is to take inspiration from how curriculum is designed for human learning into a taxonomy of subjects and use the same to prompt an off-the-shelf LLM to create data. GLAN does not need seed examples, or pre built-taxonomy like prior work. Human verification is also performed post the building of taxonomy to weed out unimportant or inaccurate divisions. The overall process is High level taxonomy -> subjects -> syllabus -> instructions.

**Strengths:**

1. Overall strong Performance: Extensive experiments show GLAN's effectiveness in various tasks, outperforming or matching state-of-the-art models in several benchmarks (Table 1)
2. Figure 2 on scaling properties of GLAN: I found this figure quite interesting. It suggests a log linear scaling trend in performance as GLAN data is scaled up. This is quite promising.
3. Section 3.5 on Task-specific overfitting: Another great analysis section that discusses how GLAN does not particularly overfit to the training data. This ensures that the synthetic data remains generalizable across different domains.
4. Modularity of the pipeline: The modular nature of the GLAN pipeline allows for easy customization and extension by incorporating new nodes into the taxonomy without re-generating the entire dataset.

**Weaknesses:**

1. No use of actual human curriculum: The paper set the expectation right in the abstract of using/getting strongly inspired from human curriculum. I was disappointed that the method does not utilize existing human curriculum structures, potentially missing out on years of insights in developing the same. Generating synthetic data, and in this case entire taxonomies from pre-existsing models can lead to extremely large amounts of bias. I would have much rather seen the authors delegate only lower level questions to LLMs than high level abstractions, which would lead to a trickle down effect on every single node in the taxonomy. This study, in my opinion, is incomplete without using either human generated taxonomies, and/or a comparison between how different the taxonomies are.
2. Computation cost not compared: The paper does not provide a comparison of computational costs with similar methods, such as WizardLM. For instance, GLAN training required approximately 8 days using 32 A100 GPUs to generate 10 million instructions, but no direct comparisons are made to illustrate the efficiency or cost-effectiveness relative to other approaches.
3. The method is limited by the performance of GPT-3.5/4: The quality of the generated taxonomy and syllabus heavily depends on the capabilities of the underlying LLMs used in the process, namely GPT-3.5 and GPT-4. In general, GLAN does not inform how we can improve capabilities of models beyond GPT4. But also, does not consider the cost of generating 10 million instructions.
4. High variability in results (Table 2): There is significant variability in GLAN's performance across different categories, with particularly weaker results in humanities and social sciences compared to STEM fields. The authors should address this, also discuss the document proportion of each taxonomy, and potentially see if there is a correlation between the data size and performance.

**Questions:**

**Did you perform an ablation to verify this hypothesis?**: "For instance, the subject calculus is both in physics and mathematics. We do not de-duplicate those subjects, since it may reflect their importance in human knowledge."

Please see Weaknesses.

**Limitations:**

Please see Weaknesses

---

> ### Author Rebuttal · Authors · 2024-08-07
>
> We appreciate your detailed review and the valuable suggestions provided for improving our work.
>
> ---
>
> > No use of actual human curriculum
>
> We did not use human curriculum explicitly to build our taxonomy because:
>
> 1) Initially, we intend to automate the whole generation process and found GPT-4 is good at listing the taxonomy structure of human curriculum.
>
> 2) We have a verification process for the taxonomy (see Section 2.1 and Section 3.1). Actually, all of our annotators searched for existing human curriculum taxonomies for verification. That said, we actually leverage human curriculum implicitly and our taxonomy overlaps significantly with existing human curriculum.
>
> 3) There are numerous different versions of the human curriculum on the web and determining which version to use is challenging. We instead generate the taxonomy using GPT-4 (we assume GPT-4 has read all these versions and can produce a version with above average quality) and then verify each node inside it.
>
> We also did not use human created syllabi. Because the syllabi developed within human curricula are numerous, and collecting them all to ensure high-quality and comprehensiveness is impractical for us.
>
> We will add these discussions to our updated manuscript.
>
> ---
>
> > Computation cost not compared: The paper does not provide a comparison of computational costs with similar methods, such as WizardLM. For instance, GLAN training required approximately 8 days using 32 A100 GPUs to generate 10 million instructions, but no direct comparisons are made to illustrate the efficiency or cost-effectiveness relative to other approaches.
>
> For the **computational cost, API cost and GLAN’s differences from other approaches**, please refer to the **Author Rebuttal** addressed to all reviewers.
>
> ---
>
> > The method is limited by the performance of GPT-3.5/4: The quality of the generated taxonomy and syllabus heavily depends on the capabilities of the underlying LLMs used in the process, namely GPT-3.5 and GPT-4. In general, GLAN does not inform how we can improve capabilities of models beyond GPT4. But also, does not consider the cost of generating 10 million instructions.
>
> - **Model Dependency**: GLAN is not limited to GPTs. It can be applied to any strong close-source (e.g., Claude, Gemini) or open-source (e.g., Llama-3.1 70B/405B, Nemotron-4, Mistral Large 2, etc.) LLMs.. We chose GPT-3.5/4 for our experiments to best demonstrate the effectiveness of our method at the time of writing.
>
> - **Improving Model Capabilities Beyond GPT-4**: This is still an open research problem, focusing on how a language model can self-improve with its own generations. Solely leveraging GLAN may not directly enhance capabilities beyond those of GPT-4. But we believe the diverse instructions GLAN can produce across a wide range of domains and tasks can at least address the question of “where to improve in model self-improving?”
>
> - **Cost of Generating Instructions**: Please refer to our **Author Rebuttal** at the top.
>
> ---
>
> > High variability in results (Table 2): There is significant variability in GLAN's performance across different categories, with particularly weaker results in humanities and social sciences compared to STEM fields. The authors should address this, also discuss the document proportion of each taxonomy, and potentially see if there is a correlation between the data size and performance.
>
>
> - **High variability in results**: We carefully examined the disciplines in our taxonomy (esp. for those related to MMLU subjects). We found there are 19 disciplines related to MMLU and only 6 of them are STEM disciplines. In our final experiment, we generated almost the same number of examples for each of these 19 disciplines. Consequently, we have more non-STEM data than STEM data.
> Therefore, as mentioned in line 264 to 269, the strong STEM results may be due to CoT reasoning.
>
> - **Ablation on duplicated subjects**: We did not do ablation on duplicated subjects for the following reasons. 1) The duplication is not very severe in our view. We have in total 15,751 subjects and 7,030 of them are unique. Since we repeat the subject generation for each discipline for 10 times (described in line 192), the duplication of subjects here is reasonable. 2) We manually inspected the most frequent subjects and they looked reasonable for us.
>
> ---
>
> Thanks again for your review! All the discussions and analysis above will be added to our updated manuscript.

---

> > ### Author Response · Authors · 2024-08-12
> >
> > Dear Reviewer 7Qp1,
> >
> > Since we are approaching the deadline of the discussion period, we were wondering if you have had the chance to review our response to your comments. We would like to kindly inquire about the extent to which we have successfully addressed the concerns outlined in your review. We greatly value your feedback and would appreciate any further questions or comments you might have.
> >
> > Thank you for your time and consideration.
> >
> >
> > Sincerely,
> >
> > All Authors

---

> ### Comment · Reviewer_7Qp1 · 2024-08-13
>
> > Initially, we intend to automate the whole generation process and found GPT-4 is good at listing the taxonomy structure of human curriculum.
>
> "found GPT-4 is good at listing the taxonomy" : this is based on what metric? was there a scientific study to test this?
>
> > our taxonomy overlaps significantly with existing human curriculum.
>
>
> Is there a scientific analysis of this claim?
>
> > computation cost comparison.
>
>
> I was looking for a training compute cost comparison since all baselines may have been trained for different durations, which can be a big confounder in the "perceived quality" of the data.
>
> Overall, found the answer to "high variability" quite unconvincing, and a reluctance to scientifically examine the phenomenon that is underneath the variability (data density etc.).

---

> > ### Author Response · Authors · 2024-08-14
> >
> > Thank you for your thoughtful and detailed feedback. We greatly appreciate the time and effort you have taken to engage with our work and provide valuable insights.
> >
> > > a more scientific analysis of "found GPT-4 is good at listing the taxonomy structure" and "our taxonomy overlaps significantly with existing human curriculum"
> >
> > To quantify the overlap, we calculated the Overlap Coefficient (Szymkiewicz-Simpson coefficient; refer to Equation below) between our taxonomy and a "standard curriculum" (https://en.wikipedia.org/wiki/Outline_of_academic_disciplines)
> > $$O(A, B) = \frac{|A \cap B|}{\min(|A|, |B|)}$$
> > the Overlap Coefficient is 90 / 126 = 71.42%
> >
> > This high overlap coefficient supports our claim that “our taxonomy overlaps significantly with existing human curriculum” and is also a strong indicator that “GPT-4 is good at listing the taxonomy structure of human curriculum”. We also find that recent studies [1][2] have demonstrated strong capabilities of GPT-4’s in generating taxonomies.
> >
> > We have uploaded our taxonomy as supplementary material (please refer to line 186 in Section 3.1). Besides, if you are aware of a more appropriate standard curriculum for comparison, we would appreciate your suggestions.
> >
> > > computation cost comparison.
> >
> > Training cost (measured in FLOPs) for different methods are shown below
> > | **Methods**       | **FLOPs**         |
> > |-------------------|-------------------|
> > | Orca 2 [3]        | $$9.5 \times 10^{20}$$ |
> > | MetaMath [5]         | $$2.5 \times 10^{19}$$ |
> > | WizardLM v1.2 [6]    | $$\geq 3.2 \times 10^{19}$$ |
> > | WizardMath v1.1 [7]  |   –            |
> > | GLAN              | $$6.3 \times 10^{20}$$ |
> >
> > According to [6], the training cost for WizardLM v1.0 is $3.2 \times 10^{19}$ FLOPs, while the training cost for WizardLM v1.2 remains unknown, though we expect that at least the same number of examples were used. The exact number of training examples for WizardMath is not disclosed in [7], and it also involves a PPO training stage with limited technical details provided. Please refer to Table 1 for their performance comparisons. Despite utilizing less computational resources, our results surpass those of Orca 2, and we consistently outperform MetaMath and WizardLM across all tasks presented in Table 1.
> >
> > Also note that GLAN data aims to enhance a model's capabilities across a wide range of tasks (without relying on seed examples), whereas the data generated by previous methods such as MetaMath, WizardLM and WizardMath are focused on improving performance in specific tasks. Therefore, to achieve the same level of performance on a particular task, our method usually needs to generate more data and hence higher training cost.
> >
> > > Overall, found the answer to "high variability" quite unconvincing, and a reluctance to scientifically examine the phenomenon that is underneath the variability (data density, etc.).
> >
> > High variability  in humanities and social sciences performance is not due to data density but the inference strategy (CoT vs. non-CoT).
> >
> > We observed no correlation between the data size and performance. Specifically, there are 19 disciplines related to MMLU subjects and only 6 of them are STEM disciplines. We generated almost the same number of examples for each of these 19 disciplines. We have more non-STEM data than STEM data with the non-STEM data being 2.16 times greater.
> >
> > We believe that the Chain-of-Thought (CoT) reasoning contributes to the lower performance on these questions. In our earlier experiments using the MMLU benchmark, we evaluated both CoT and non-CoT settings. While the overall performance was superior with CoT (leading to its adoption), we noticed that CoT was more effective for STEM questions, whereas non-CoT proved advantageous for humanities and social sciences questions. This might be because CoT aids in multi-step reasoning in STEM multiple-choice questions, while humanities and social sciences questions involve more memorization and single-step reasoning, where CoT may introduce additional errors (also see line 264 to 269).
> >
> >
> >
> > All discussions and results will be added to our revised manuscript.
> >
> > Thank you again for your valuable feedback.
> >
> > ## References
> >
> > [1]. Gunn, M. et al. “Creating a Fine Grained Entity Type Taxonomy Using LLMs.” 2024.
> >
> > [2]. Lee, M. et al. “Human-AI Collaborative Taxonomy Construction: A Case Study in Profession-Specific Writing Assistants.” 2024.
> >
> > [3]. Mitra, A. et al. “Orca 2: Teaching Small Language Models How to Reason.” 2023.
> >
> > [4]. Stanford Alpaca GitHub Repository
> >
> > [5]. Yu, L. et al. “MetaMath: Bootstrap Your Own Mathematical Questions for Large Language Models.” ICLR 2024.
> >
> > [6]. C. Xu et al. “WizardLM: Empowering Large Language Models to Follow Complex Instructions”. 2023
> >
> > [7]. H. Luo et al. “WizardMath: Empowering Mathematical Reasoning for Large Language Models via Reinforced Evol-Instruct”. 2023

---

### Official Review · Reviewer_VaBs · 2024-07-31

**Soundness:** 3
**Presentation:** 4
**Contribution:** 3
**Rating:** 6
**Confidence:** 3

**Summary:**

## Overall summary
- This paper introduces GLAN, a method for enhancing LLMs by generating synthetic instruction data using a taxonomy of human knowledge and capabilities. GLAN constructs this taxonomy by decomposing knowledge into fields and disciplines, leveraging LLMs for generating a comprehensive syllabus for each subject.
- GLAN’s scalable and customizable framework allows for easy integration of new fields of skills, highlighting its potential for ongoing improvement and adaptation.

## My opinion of the paper
- I think this is a really interesting approach to generate data that can allow LLMs to be potentially smarter. However, I am wondering if there are newer topics, for example (within the medical area, we have the new topic called "Covid-19".) Since GLAN is very dependent on LLMs, the main area of concern would be ensuring that the LLMs that GLAN depends on remains updated.

**Strengths:**

## Originality
- The approach is quite interesting. The authors made use of real life scenarios, which is to use the structure of human education systems to build the taxonomy. This approach mimics the systematic acquisition of knowledge and skills in education, providing a framework for generating instruction data.
## Clarity
- Pseudo Algorithm provided and figures are easy to understand.
## Significance
- By creating a general and scalable method for instruction tuning, GLAN has the potential to improve the performance of LLMs across a wide range of tasks and domains.

**Weaknesses:**

## Quality
- While the paper claims scalability, there is limited discussion on the computational resources required for generating the synthetic data at scale. Practical constraints related to computational costs and time could be a potential weakness. It was mentioned in the checklist that it is very computationally expensive to repeat experiments.

**Questions:**

- Line 269: Why do you say that the reason why errors coming from humanities and social science questions is due to CoT? Could it be due to the lack of knowledge, because it was not trained on that knowledge?
- Figure 2: Any reason why for HumanEval and BBH datasets, the scores dropped even though the GLAN data size increase?

**Limitations:**

Indicated in the appendix (do consider placing it in main paper), but did not mention about computation cost like what was mentioned in the checklist.

---

> ### Author Rebuttal · Authors · 2024-08-07
>
> Thank you for your careful review and the thoughtful comments on our submission.
>
> ---
> > However, I am wondering if there are newer topics, for example (within the medical area, we have the new topic called "Covid-19".) Since GLAN is very dependent on LLMs, the main area of concern would be ensuring that the LLMs that GLAN depends on remains updated.
>
> Actually we find in our generated dataset there are 38,749 instructions and 28 syllabus containing the keyword “covid-19”. We believe the GPT-4 we used in experiments is aware of the Covid-19 topic. However, there is no syllabus or discipline exactly focusing on “Covid-19”. A quick fix is to instruct newer GPT-4 (i.e., gpt-4o) using prompts described in Section 2.2 and 2.3 (also see Table 6) to generate a syllabus or subject list on “covid-19”. The generated syllabus looks good to us and we show part of it in the following due to space limitation.
>
> ```
> ### College Level Syllabus on COVID-19
> #### **Introduction**
> This course provides an in-depth study of COVID-19 …
> #### **Class Details**
> **Class 1: Introduction to Coronaviruses**
> - **Description**: Overview of coronaviruses, …
> - **Knowledge Points**:
>   1. Structure and classification of coronaviruses.
>   2. History of coronavirus outbreaks.
>   3. Transmission mechanisms.
>   4. Symptoms and disease progression.
>   5. Comparison with other respiratory viruses.
> - **Learning Outcomes & Activities**:
>   - Understand the basic virology of coronaviruses.
>   - Activity: Research and present a comparison between SARS, MERS, and COVID-19.
>
> ---
> **Class 2: Virology of SARS-CoV-2**
> …
> **Class 3: Epidemiology of COVID-19**
> …
> **Class 4: Clinical Presentation and Diagnosis**
> …
> ```
>
> Thanks to the “easy customization” feature of GLAN (line 74 to 77), we only need to generate instructions for the newly added “covid-19” topic without re-generating the entire dataset.
>
> ---
>
> > While the paper claims scalability, there is limited discussion on the computational resources required for generating the synthetic data at scale. Practical constraints related to computational costs and time could be a potential weakness. It was mentioned in the checklist that it is very computationally expensive to repeat experiments.
>
> For the **computational cost** and **API cost**,  please refer to the **Author Rebuttal** at the top.
>
> Besides, we believe there are two types of scalability: (1) the ability to generate a large number of diverse data points and (2) the ability to generate each data point at a low cost.
> In Checklist 7 (Experiment Statistical Significance), we noted, “We did not include error bars in the experiments due to the high computational demands.” This statement does not contradict GLAN's scalability. This is because generating an additional 10 million data points to compute error bars is indeed computationally expensive. On the contrary, the scalability of GLAN is precisely why it is challenging to repeat the experiments. Our objective is to enhance the general capabilities of large language models (LLMs). Achieving this goal appears to necessitate a large dataset, which in turn results in high computational/API costs.
>
> ---
>
> > Line 269: Why do you say that the reason why errors coming from humanities and social science questions is due to CoT? Could it be due to the lack of knowledge, because it was not trained on that knowledge?
>
> The lower performance on humanities and social science questions is unlikely to stem from a lack of knowledge. The scope of knowledge for the generated questions is predominantly governed by generated syllabuses. Therefore, it is implausible that GPT-4 would create syllabuses for humanities and social sciences with significantly narrower knowledge coverage compared to those in STEM disciplines.
>
> We believe CoT contributes more to the lower performance on humanities and social science questions. In our earlier experiments using the MMLU benchmark, we evaluated both w/ CoT and w/o CoT settings.While the overall performance was superior w/ CoT (thus leading to its adoption), a nuanced observation emerged: CoT was more effective for STEM questions, whereas the absence of CoT proved advantageous for humanities and social science questions. It may be because “CoT may help the multi-step reasoning in STEM multi-choice questions, while humanities and social sciences questions involve more memorization and single-step reasoning, where CoT may introduce additional errors.”
>
> ---
>
> > Figure 2: Any reason why for HumanEval and BBH datasets, the scores dropped even though the GLAN data size increase?
>
> The score drop happens within the 50k to 1M range. The main reason is probably due to the relatively small average size of data points per discipline. We have 126 disciplines in total and on average we have:
> - 2K examples per discipline for a total of 200K examples,
> - 4K examples per discipline for a total of 500K examples,
> - 8K examples per discipline for a total of 1M examples.
>
> We do observe a significant leap in performance from 1M to 10M examples on HumanEval and BBH, when data points per domain is significant enough.
>
> We will add computational/API cost related discussions or from the checklist/appendix to the main paper.
>
> ----------
>
> Thanks again for your review! All discussions and results above will be added to our revised manuscript.

---

> > ### Comment · Reviewer_VaBs · 2024-08-08
> > **Reply to Authors' Rebuttal**
> >
> > Thank you for your clarifications!
> >
> > Regarding the first point, I apologize for the confusion. Covid-19 is indeed already known to GPT-4 as its training data extends until October 2023. I was referring to scenarios involving new topics that emerged after October 2023, though I can't think of specific examples at the moment. Would such scenarios be handled as outlined in Sections 2.2 and 2.3 as you have explained here?

---

> > > ### Author Response · Authors · 2024-08-08
> > >
> > > Thank you for your insightful question and for bringing up this important point.
> > >
> > > Regarding new topics that emerged after October 2023, it is still possible to handle these scenarios with some modifications to the underlying LLM or prompts as described in Sections 2.2 and 2.3. Here are three methods to ensure our approach remains up-to-date. In the following, we refer to the prompts in Sections 2.2 and 2.3 as "GLAN prompts":
> > >
> > > - **Using Retrieval-Augmented Generation (RAG)**: You can leverage an existing LLM integrated with web search capabilities (e.g., Perplexity AI API https://docs.perplexity.ai/reference/post_chat_completions) or implement RAG using GPT-4 function calls combined with the Google Search API. In this approach, the GLAN prompts remain unchanged. The RAG model first retrieves documents relevant to the new topic and then integrates these documents with the GLAN prompts. This method is very likely to be effective because new topics are typically related to pre-existing knowledge within the LLM. With the retrieved documents, the LLM is still likely to generate a reasonable subject list or syllabus (i.e., breaking down the new topic given docs introducing it).
> > >
> > > - **Prepending Documents of the New Topic to Our Prompts**: Conduct a search for the new topic using a search API (e.g., Google Search API) and prepend the top returned documents and "Based on the context above, your task is as follows." to the GLAN prompts. It may also be necessary to prepend these documents to the instruction generation prompt in Section 2.4 to ensure that the LLM comprehends any new terms. This method is similar to the RAG approach but provides more transparency to API users regarding the integration of search results with prompts.
> > >
> > > - **Adopting Newer LLMs**: A more straightforward solution is to wait for the release of newer models (in three months or less) by leading LLM companies (e.g., OpenAI, Anthropic, Meta, Mistral) which will likely include training on documents covering the new topic. Once these models are available, they can be adopted.
> > >
> > > We appreciate your question as it tends to broaden the scope of our method. Thank you again for your valuable feedback!

---

> > > > ### Comment · Reviewer_VaBs · 2024-08-09
> > > > **Reply to Author's Comment**
> > > >
> > > > Thank you for your clarifications! I have raised the score rating from 5 to 6.

---

> > > > > ### Author Response · Authors · 2024-08-09
> > > > >
> > > > > Thank you very much for taking the time to consider our clarifications and for your updated evaluation. We greatly appreciate your thoughtful feedback throughout the review process, which has helped to strengthen our work.

---

### Author Rebuttal · Authors · 2024-08-07

We thank reviewers for your valuable feedback; in this general rebuttal, we address common concerns and questions raised.

**Regarding Computational Cost**

“8 days using 32 A100 GPUs” is the cost of fine-tuning Mistral on the 10 million instructions we generated. We do not know the computational cost of generating these 10 million instructions, as the model architectures and parameters for GPT-4 and GPT-3.5 are not disclosed.

**Regarding API Cost**

We estimate the API cost for data generation, which amounts to approximately 360K USD when using GPT-4 and GPT-3.5 (for response generation), based on data from context.ai and OpenAI API official pricing. It is important to note that a team within our organization supports these GPT API calls and we believe the actual cost is substantially lower than 360K USD. As of today, we recommend using GPT-4o and GPT-4o-mini (for response generation) to reproduce the data, reducing the cost to approximately 66K USD. This recommendation is based on our findings that GPT-4o outperforms GPT-4 in many tasks and GPT-4o-mini consistently outperforms GPT-3.5. Furthermore, by leveraging Mistral Large 2 and Mistral 8x7B, the cost can be further reduced to around 42K USD.

Notably, the API costs have decreased significantly since last year:
- GPT-4-0613: 30/60 USD per million input/output tokens
- GPT-4-Turbo-1106: 10/30 USD per million input/output tokens
- GPT-4o: 5/15 USD per million input/output tokens
- GPT-4o-2024-08-06： 2.5/10 USD per million input/output tokens
- GPT-4o-mini: 0.15/0.60 USD per million input/output tokens

We anticipate that the API costs will continue to decrease in the future, making the application of GLAN more feasible.

**Differentiating GLAN from Other Approaches**

This work does not aim to achieve SOTA results on existing benchmark tasks using minimal resources. Intentionally generating data similar to target tasks (e.g., paraphrasing an existing training set) is perhaps the most cost-effective method to improve on these target tasks. However, in the long run, this class of methods lead to overfitting on these tasks. LLMs of today (even 7B models) are capable of solving many different tasks, and existing benchmark tasks are only a small subset of them. Our method, GLAN, aims to enhance the capabilities of LLMs across a wide range of tasks (not just the tasks with good evaluations), and we do not use training data from target tasks at all. Our assumption is that all instruction data for different tasks can be generated using the same method; if we can perform well on known tasks (with good evaluations), we can probably also do well on tasks still lacking evaluations. Results in Section 3.3 and 3.6 demonstrate that we did reasonably well on existing known tasks.

In short, GLAN aims to enhance capabilities across a wide range of tasks, whereas previous methods such as WizardLM aim to enhance capabilities on one or a few tasks. To achieve the same level of performance on a particular task, our method needs to generate more data, which is the price to pay for cross-task generalization.

---

### Decision · Program_Chairs · 2024-09-25

**Decision:**

Reject

**Comment:**

The paper proposes a top-down approach to generate high-quality instruction tuning data, which covers taxonomy, subjects, syllabus and instructions. The generated taxonomy is verified by human to guarantee its quality. The performance is strong on many benchmarks such as reasoning, coding and general instruction following. Several reviewers raised concerns about the cost to generate the 10 Million instructions. And there is a mismatch between the motivation to use pre-curated taxonomy of human knowledge and the actual method that relies heavily on LLMs to establish the taxonomy.